# Amyloid Burden Correlates with Electrocardiographic Findings in Patients with Cardiac Amyloidosis—Insights from Histology and Cardiac Magnetic Resonance Imaging

**DOI:** 10.3390/jcm13020368

**Published:** 2024-01-09

**Authors:** Franz Duca, René Rettl, Christina Kronberger, Michael Poledniczek, Christina Binder, Daniel Dalos, Matthias Koschutnik, Carolina Donà, Dietrich Beitzke, Christian Loewe, Christian Nitsche, Christian Hengstenberg, Roza Badr-Eslam, Johannes Kastner, Jutta Bergler-Klein, Andreas Anselm Kammerlander

**Affiliations:** 1Division of Cardiology, Department of Internal Medicine II, Medical University of Vienna, 1090 Vienna, Austria; franz.duca@meduniwien.ac.at (F.D.); rene.rettl@meduniwien.ac.at (R.R.); christina.kronberger@meduniwien.ac.at (C.K.); michael.poledniczek@meduniwien.ac.at (M.P.); christina.binder@meduniwien.ac.at (C.B.); daniel.dalos@meduniwien.ac.at (D.D.); matthias.koschutnik@meduniwien.ac.at (M.K.); carolina.dona@meduniwien.ac.at (C.D.); christian.nitsche@meduniwien.ac.at (C.N.); roza.badreslam@meduniwien.ac.at (R.B.-E.); johannes.kastner@meduniwien.ac.at (J.K.); jutta.bergler-klein@meduniwien.ac.at (J.B.-K.); 2Division of Cardiovascular and Interventional Radiology, Department of Bioimaging and Image-Guided Therapy, Medical University of Vienna, 1090 Vienna, Austria; dietrich.beitzke@meduniwien.ac.at (D.B.); christian.loewe@meduniwien.ac.at (C.L.)

**Keywords:** cardiac amyloidosis, transthyretin amyloidosis, light chain amyloidosis, electrocardiogram, histology, T1 mapping, extracellular volume, cardiac magnetic resonance imaging

## Abstract

Cardiac amyloidosis (CA) is associated with several distinct electrocardiographic (ECG) changes. However, the impact of amyloid depositions on ECG parameters is not well investigated. We therefore aimed to assess the correlation of amyloid burden with ECG and test the prognostic power of ECG findings on outcomes in patients with CA. Consecutive CA patients underwent ECG assessment and cardiac magnetic resonance imaging (CMR), including the quantification of extracellular volume (ECV) with T1 mapping. Moreover, seven patients underwent additional amyloid quantification using immunohistochemistry staining of endomyocardial biopsies. A total of 105 CA patients (wild-type transthyretin: 74.3%, variant transthyretin: 8.6%, light chain: 17.1%) were analyzed for this study. We detected correlations of total QRS voltage with histologically quantified amyloid burden (r = −0.780, *p* = 0.039) and ECV (r = −0.266, *p* = 0.006). In patients above the ECV median (43.9%), PR intervals were significantly longer (*p* = 0.016) and left anterior fascicular blocks were more prevalent (*p* = 0.025). In our survival analysis, neither Kaplan–Meier curves (*p* = 0.996) nor Cox regression analysis detected associations of QRS voltage with adverse patient outcomes (hazard ratio: 0.995, *p* = 0.265). The present study demonstrated that an increased amyloid burden is associated with lower voltages in CA patients. However, baseline ECG findings, including QRS voltage, were not associated with adverse outcomes.

## 1. Introduction

In recent years, cardiac amyloidosis (CA) has become an increasingly acknowledged heart failure (HF) etiology, particularly since the approval of amyloid-specific drugs [1,2,3]. The pathophysiological hallmark of CA is the deposition of amyloid within the myocardial extracellular space [4].

Wild-type transthyretin (ATTRwt) amyloidosis is by far the most prevalent form. This disease is strongly age-associated and was therefore formerly known as senile amyloidosis. ATTRwt CA can be found in up to 25% of the general population aged ≥85 years, in 13% of patients affected by HF with preserved ejection fraction (HFpEF), and in up to 16% of patients who undergo transcatheter aortic valve replacement (TAVR) [5,6,7]. However, ATTR CA can also be caused by specific point mutations in the transthyretin (TTR) gene and is then referred to as variant ATTR (ATTRv) amyloidosis. This type of ATTR CA is, except for certain endemic areas like northern Portugal, indeed a rare disease. AL CA is the second most common form of CA and develops due to the deposition of misfolded immunoglobulin light chains produced by a plasma cell clone [8].

Irrespective of its precursor protein (ATTR versus AL), affected patients feature several distinct electrocardiographic (ECG) patterns. The most prominent features, which are also considered red flags in the diagnostic work-up of CA patients, are low(er) voltage and a so-called pseudoinfarct pattern [9]. Furthermore, CA patients can present with a plethora of arrhythmias, of which atrial fibrillation (A-Fib) is most prevalent [10].

Despite efforts to investigate potential contributing factors to CA ECG features, studies investigating how the amount of amyloid deposited in the myocardial extracellular space is associated with these ECG alterations are lacking [11].

We therefore performed detailed analyses of CA ECGs and correlated findings with amyloid burden as well as structural and functional cardiac parameters using histological analysis of endomyocardial biopsies (EMB) and cardiac magnetic resonance (CMR) imaging.

## 2. Methods

### 2.1. Setting and Study Design

The present study was conducted at the Department of Cardiology of the Medical University of Vienna within the scope of a prospective single-center patient registry. This study was carried out in accordance with good clinical practice guidelines as outlined in the declaration of Helsinki and approved by the ethics committee of the Medical University of Vienna (EK#2036/2015). Moreover, written informed consent was obtained from all patients before enrollment.

Key inclusion criteria for the present study, besides a diagnosis of CA, were the ability to undergo CMR imaging with extracellular volume (ECV) quantification and ECG. Exclusion criteria were the presence of pericardial effusion and morbid obesity due to their known influence on QRS voltage [12].

### 2.2. Diagnosis of Cardiac Transthyretin Amyloidosis

After the publication of the non-invasive diagnostic algorithm by Gillmore et al. in 2016, diagnosis of cardiac ATTR amyloidosis was made if patients had Perugini grade ≥2 myocardial tracer uptake on bone scintigraphy and if the presence of a paraprotein was ruled out [13]. In patients with ambiguous non-invasive test results and before 2016, ATTR amyloidosis was diagnosed if EMB samples stained positive for Congo red, showed apple green birefringence under polarized light, and reacted with anti-ATTR antibodies. Gene sequencing was offered to all patients diagnosed with ATTR CA.

### 2.3. Diagnosis of Cardiac Light Chain Amyloidosis

Cardiac AL amyloidosis was diagnosed if biopsy samples showed Congo red positivity, apple green birefringence under polarized light, and reactivity with AL antibodies. In cases of extra-cardiac biopsy samples, cardiac involvement was defined via mean left ventricular (LV) wall thickness (septum and posterior wall) > 12 mm in the absence of hypertension or other potential causes of LV hypertrophy in accordance with current recommendations from the International Symposium on Amyloid and Amyloidosis [14].

### 2.4. Electrocardiogram

ECGs were recorded at baseline in a supine position with lead V1 placed in the 4th intercostal space at the right sternal border, V2 in the 4th intercostal space at the left sternal border, V3 midway between leads V2 and V4, V4 in the 5th intercostal space in the mid-clavicular line, V5 in the 5th intercostal space in the anterior axillary line, and V6 in the 5th intercostal space in the mid-axillary line. Limb leads were placed at the wrists and ankles. The recording speed was 25 mm/s and amplification was 10 mm/mV. All ECGs were manually analyzed by board-certified cardiologists who were blinded to patient outcomes.

ECG parameters of interest were rhythm (sinus rhythm versus A-fib versus atrial flutter (A-flu)), heart rate, cardiac vector, PR interval (ms), AV blocks (I°–III°), bundle branch blocks (complete, incomplete, right bundle branch block (RBBB), left bundle branch block (LBBB), anterior fascicular block, posterior fascicular block, bi-fascicular block, tri-fascicular block), QRS width (ms), heart rate-corrected QT time (ms), presence of ventricular ectopic beats, anterior pseudoinfarction pattern, low QRS voltage (QRS amplitude in each precordial lead <10 mm (<1.0 mV), or <5 mm (<0.5 mV) in all peripheral leads), and total as well as precordial and limb QRS voltage (1 mm/0.1 mV) [11,15].

### 2.5. Cardiac Magnetic Resonance Imaging

All cardiovascular MRI studies were performed on a 1.5-T system (Avanto FIT; Siemens Medical Solutions, Erlangen, Germany), including late gadolinium enhancement (LGE) imaging. For cine imaging, steady-state free precision images were used (repetition time ms/echo time ms, 3.2/1.2; flip angle, 64°; voxel size, 1.4 × 1.4 × 6 mm; matrix, 180 × 256 pixels). For late gadolinium chelate enhancement imaging, segmented inversion recovery sequences (700/1.22; flip angle, 50°; voxel size, 1.4 × 1.4 × 8 mm; 146 × 256 matrix) were performed at least 10 min after injection of 0.1 mmol/mL gadobutrol (Gadovist; Bayer Vital GmbH, Leverkusen, Germany). T1 mapping was performed with electrocardiographically triggered MOLLI with a 5(3)3 prototype (5 acquisition heartbeats followed by 3 recovery heartbeats and a further 3 acquisition heartbeats) on a short-axis mid-cavity slice and with a 4-chamber view. T1 sequence parameters were as follows: starting inversion time (TI) 120 ms, TI increment 80 ms, reconstructed matrix size 256 × 218, measured matrix size 256 × 144 (phase-encoding resolution 66%, phase-encoding field of view 85%). T1 maps were created both before and 15 min after contrast agent application. For post-contrast T1 mapping, a 4(1)3(1)2 prototype was used. To counteract the patchy distribution of amyloid within the myocardium, regions of interest (ROIs) in our T1 maps included the whole myocardium (excluding the endocardium) in mid-cavity short-axis views and four-chamber views. T1 times from ROIs were averaged for ECV calculation, which was performed as previously described [16]. In all patients, venous blood for conventional hematocrit measurement was drawn when placing the intravenous line for contrast agent administration.

### 2.6. Endomyocardial Biopsy and Histological Analysis

EMB specimens were acquired from the LV during left heart catheterization (Bipal ^®^ biopsy forceps, Cordis^®^ Corporation, Bridgewater, NJ, USA). Specimens were immediately fixed in 7.5% buffered formalin for 24 h followed by paraffin embedding. EMB samples were cut in 2 µm and 6 µm sections using a Leica RM 2255 Microtome (Charleston, SC, USA). Of these, the 6 µm sections were used for Congo red staining and visualization under polarized light and the 2 µm slices were used for immunohistochemical staining (AmY-kit amyloid antibodies, Martinsried, Germany). After staining, slides were scanned at 20-fold magnification with a Zeiss Observer Z1 microscope (Carl Zeiss Microsocopy GmbH, Jena, Germany) and TissueFAXS software (TissueGnostics^®^ version 7.1.123, Vienna, Austria). HistoQuest software (TissueGnostics^®^, Vienna, Austria) version 7.1.1.129 was used for amyloid quantification. Then quantification of the amyloid area in the scanned EMB samples was performed with ImageJ software using a color threshold macro based on an algorithm by G. Landini (version 1.47, downloaded on 8 July 2013, available at: (http://www.mecourse.com/landinig/software/software.html accessed on 1 August 2014). The mean amyloid area percentage of the available biopsy specimens (*n* = 3–5) was used for statistical analyses.

### 2.7. Endpoints and Outcome Measures

The co-primary endpoints of our study were correlations of histologically and non-invasively quantified (ECV) amyloid burden with ECG parameters. Furthermore, we investigated the prognostic power of ECG parameters in patients with CA. For outcome analyses, a composite of all-cause death, heart transplantation, and cardiovascular-related hospitalizations was chosen. Outcome events were documented during follow-up at our outpatient clinic, during telephone visits, or retrieved from electronic medical records as well as the Austrian death registry.

### 2.8. Statistical Analysis

Continuous variables are expressed as medians and interquartile ranges (IQRs). Categorical variables are presented as numbers and percentages. Continuous variables were compared using the Mann–Whitney U test. Pearson and Spearman correlation coefficients as well as linear regression models were used to report the relationship between total ECG voltage and clinical, laboratory, ECG, as well as CMR parameters. Kaplan–Meier curves with respective log-rank tests and Cox regression models were used for outcome analyses.

IBM SPSS© version 27.0 (IBM Corp. Chicago, IL, USA) and STATA© 16 (StataCorp. College City, CA, USA) were used for statistical analysis. A *p* value of ≤0.05 was set as the level of significance.

## 3. Results

### 3.1. Study Population

Between February 2012 and November 2021, a total of 105 CA patients were consecutively recruited at our amyloidosis outpatient clinic and included in the present analysis.

Among our study cohort, 40.0% (*n* = 42) were referred to our outpatient clinic via internal hospital referrals and 41.0% (*n* = 43) via external referrals. Of note, for 19.0% (*n* = 20), the mode of referral was unknown. The majority of internally referred patients came from the cardiology outpatient clinic (*n* = 21), followed by the hematooncology outpatient clinic (*n* = 11). The remainder of this cohort came via cardiology wards (*n* = 4), the nephrology outpatient clinic (*n* = 3), the department of radiology (*n* = 1), the pulmonology ward (*n* = 1), and the cardiac surgery outpatient clinic (*n* = 1).

External referrers were cardiology outpatient clinics (*n* = 19), outpatient cardiologists (*n* = 9), internal medicine outpatient clinics (*n* = 7), outpatient internal medicine specialists (*n* = 5), cardiology wards (*n* = 2), and a general practitioner (*n* = 1).

### 3.2. Baseline Characteristics

Baseline patient characteristics are depicted in Table 1. The study population at hand represents a contemporary CA cohort with 74.3% wild-type ATTR, 8.6% variant ATTR (ATTRv), and 17.1% AL CA patients. The following mutations were found among our ATTRv cohort: p.His108Arg (*n* = 5), p.Ile127Val (*n* = 1), p.Val140Ile (*n* = 1), p.Val113Leu (*n* = 1), and p.Thr69Ile (*n* = 1). The median age of our cohort was 77 years and 74.3% were male. Around 50% of our patients were in New York Heart Association (NYHA) class ≥3 and median N-terminal pro brain natriuretic peptide (NT-proBNP) was 2020 pg/mL.

Among ECG parameters, patients had a total voltage sum of 102 mm [interquartile range (IQR): 81.5–123]. Peripheral lead voltage and precordial lead voltage sums were 34.0 mm (IQR: 23.8–41.8) and 67.0 mm (IQR: 54.3–78.5), respectively. Low voltage, defined as QRS amplitude in each precordial lead <10mm or <5 mm in all peripheral leads, was present in 27.6% of our study cohort. Almost two-thirds were in sinus rhythm (62.9), while 34.3 were in A-fib and 2.9% were in A-flu. AV block 1° was found in 17.1%. Bundle blocks were common in our cohort, with complete left bundle branch blocks (LBBBs) being the most prevalent (16.2%). An anterior pseudoinfarct pattern was present in 20.0%. Furthermore, 7.6% of our patients displayed ventricular premature complexes.

With respect to CMR characteristics, our patient cohort featured CA-typical alterations such as an increased ECV (43.9%), significant myocardial hypertrophy (intraventricular septum thickness: 18.0 mm), and a preserved left ventricular ejection fraction ((LVEF) 54.3%).

### 3.3. Correlation and Regression Analyses

With the exception of histologically quantified amyloid area (r = −0.780, *p* = 0.039), we could detect weak but statistically significant negative correlations of total QRS voltage with troponin t (r = −0.221, *p* = 0.028) and ECV (r = −0.266, *p* = 0.006) (Table 2, Figure 1A,B). Moreover, QRS voltage correlated weakly with QRS width (r = 0.212, *p* = 0.030), interventricular septum (IVS), r = 0.246, *p* = 0.011), and left atrial area (r = 0.229, *p* = 0.019) (Table 2).

In our univariable regression analysis (Table 3), we found associations of QRS voltage with troponin t (Beta: −0.136, 95% confidence interval (CI): −0.257–−0.015, *p* = 0.028), QRS width (Beta: 0.218, 95% CI: 0.045–0.391, *p* = 0.014), histologically quantified amyloid area (Beta: −0.913, 95% CI: −1.756–−0.070, *p* = 0.039), ECV (Beta: −0.530, −0.939–−0.122, *p* = 0.011), IVS (Beta: 1.901, 95% CI: 0.637–3.165, *p* = 0.004), and left atrial (LA) area (Beta. 0.767, 95% CI: 0.128–1.405, *p* = 0.019).

When adjusted for type of amyloid (ATTR/AL), sex, as well as body mass index (BMI), ECV (adj. Beta: −0.610, 95% CI: −1.017–−0.204, *p* = 0.004), QRS width (adj. Beta: 0.227, 95% CI: 0.048–0.406, *p* = 0.013), IVS (adj. Beta: 1.749, 95% CI: 0.403–3.096, *p* = 0.011), LA area (adj. Beta: 0.793, 95% CI: 0.160–1.426, *p* = 0.015), and LV mass (adj. Beta: 0.129, 95% CI: 0.002–0.256, *p* = 0.046) showed statistically significant associations with total QRS voltage.

### 3.4. Electrocardiographic Parameters Stratified According to Extracellular Volume Median

When stratified according to median ECV (43.9%), PR interval was significantly longer (180 ms versus 204 ms, *p* = 0.016) and left anterior fascicular block was more prevalent (1.9% versus 13.5%, *p* = 0.025) in patients with higher ECV values (Table 4). Further, however, statistically non-significant differences were higher prevalences of low voltage (22.6% versus 32.7%), AV-Blocks (13.2% versus 23.1%), and atrial fibrillation (26.4% versus 42.3%) in the cohort above the ECV median.

### 3.5. Survival Analysis

Neither Kaplan–Meier curves and the respective log-rank test (Figure 2A,B, *p* = 0.837 and *p* = 0.956) nor Cox regression analysis showed an association of QRS voltage with adverse patient outcomes (Appendix A, hazard ratio: 0.995, 95% CI: 0.987–1.004, *p* = 0.265).

## 4. Discussion

In our prospective study of 105 patients with CA, we detected that lower QRS voltages are associated with higher amounts of histologically and non-invasively quantified amyloid within the myocardium (graphical abstract). Moreover, patients with higher ECV show further ECG alterations, such as prolonged PR intervals and higher prevalences of left anterior fascicular blocks. However, in contrast to CMR ECV, ECG parameters were not associated with adverse patient outcomes.

### 4.1. Correlation of Amyloid Burden with Electrocardiographic Findings in Cardiac Amyloidosis

Cipriani and colleagues thoroughly investigated clinical correlates and the prognostic value of QRS voltage, an ECG feature relatively prevalent in CA, and hypothesized that amyloid infiltration could be one of the main drivers behind this ECG feature. However, correlations of ECG parameters with non-invasively and/or invasively quantified amyloid were lacking. The present study is the first to demonstrate that immunohistochemically quantified amyloid burden indeed shows a strong correlation with QRS voltage (*n* = 7, R = −0.780). Moreover, QRS voltage also correlated with ECV (*n* = 105, R = −0.246). Furthermore, we detected correlations of ECV with PQ interval. This finding could be explained with the fact that histopathological studies found extensive atrial and basal ventricular amyloid infiltration, anatomic regions which are in close proximity to the AV node [17,18].

### 4.2. Prevalence of Electrocardiographic Findings in Cardiac Amyloidosis

In line with earlier studies, the median total QRS voltage sum was significantly lower when compared to published normal reference values (102 mm versus 129 mm) [11,19,20]. However, low voltage as defined by a QRS amplitude in each precordial lead <10 mm (<1.0 mV) or <5 mm (<0.5 mV) in all peripheral leads was present only in 27.6% of our CA patients. This supports the shift in paradigm that low voltage, when strict definitions are applied, is not as prevalent as previously thought [9,21]. Importantly, our study excluded patients with pericardial effusions, a condition associated with low QRS voltage and frequently observed in patients with CA [22].

Another frequently reported ECG finding in patients with CA is the anterior pseudoinfarct pattern which is, according to the literature, present in 23% to 69% of CA patients [9,23]. In our study cohort, a pseudoinfarct pattern was found in 20.0% of patients. A possible explanation for its relatively low prevalence could be that patients were in less advanced disease stages compared to other cohorts with higher prevalences [9,24].

### 4.3. Prognostic Significance of Electrocardiographic Findings in Cardiac Amyloidosis

Interestingly, despite the fact that ECV and histologically quantified amyloid are associated with adverse patient outcomes and correlated with QRS voltage, neither QRS voltage nor low voltage were associated with the combined endpoint of all-cause death, heart transplantation, or cardiovascular-related hospitalization in our study cohort [25]. This is in contrast to early studies by Kristen et al. and Sperry et al. [26,27]. However, both of these studies applied different and clinically less often used low voltage criteria [11]. Moreover, these studies did not exclude patients with pericardial effusions or cardiac devices, therefore potentially including patients in more advanced disease stages. Also, the endpoints differed, as we used similar endpoints as the landmark phase III trials (ATTR-ACT and ATTRibute-CM), while the aforementioned studies used all-cause death alone [2,28]. In a trial by Cipriani et al., low QRS voltage was also predictive of patient outcomes [11]. Again, the primary endpoint differed from our study cohort, as the authors chose cardiovascular death as the primary endpoint. Another explanation could be that in comparison to their study, our cohort was significantly smaller (*n* = 105 versus *n* = 411) and patients with pericardial effusions or cardiac devices were not excluded.

The presence of a pseudoinfarct pattern was not associated with an adverse outcome in our cohort but was reported to be a prognostic factor in a study by Zhao et al. [24]. Compared to our study population, they investigated only AL CA patients in advanced disease stages.

## 5. Limitations

The major limitations of our study are the relatively small sample size of our cohort, especially the number of patients undergoing EMB, as well as the single-center design of our study. Nonetheless, our study is the first to prospectively correlate invasively quantified amyloid with ECG findings and adds a significant number of patients to the existing literature. Furthermore, limiting data collection to a single center has the advantages of constant quality of work-up, adherence to a constant clinical routine, and constant follow-up. The generalizability of our study might be limited due to the exclusion of patients with cardiac devices, pericardial effusions, BMI ≥ 35 kg/m^2^, and potential selection bias towards CA patients with more pronounced ECG findings. However, our main focus was to investigate the influence of amyloid deposition/ECV on ECG findings and the aforementioned conditions are known to either influence CMR imaging and/or ECGs.

## 6. Conclusions

The present results support the hypothesis that myocardial amyloid depositions are responsible for lower voltages in CA patients. However, higher ECV values are not necessarily associated with a higher prevalence of abnormal ECG findings, suggesting additional pathophysiological mechanisms behind typical ECG findings in patients with CA. Moreover, baseline ECG findings were not associated with adverse outcomes. Of note, the results of our study are of limited generalizability due to its relatively small sample size and potential selection bias due to the exclusion of patients with pericardial effusion, implantable cardiac devices, and BMI ≥ 35 kg/m^2^. Therefore, further studies are needed to investigate the prognostic significance of ECG changes during the disease course of CA patients.

## Figures and Tables

**Figure 1 jcm-13-00368-f001:**
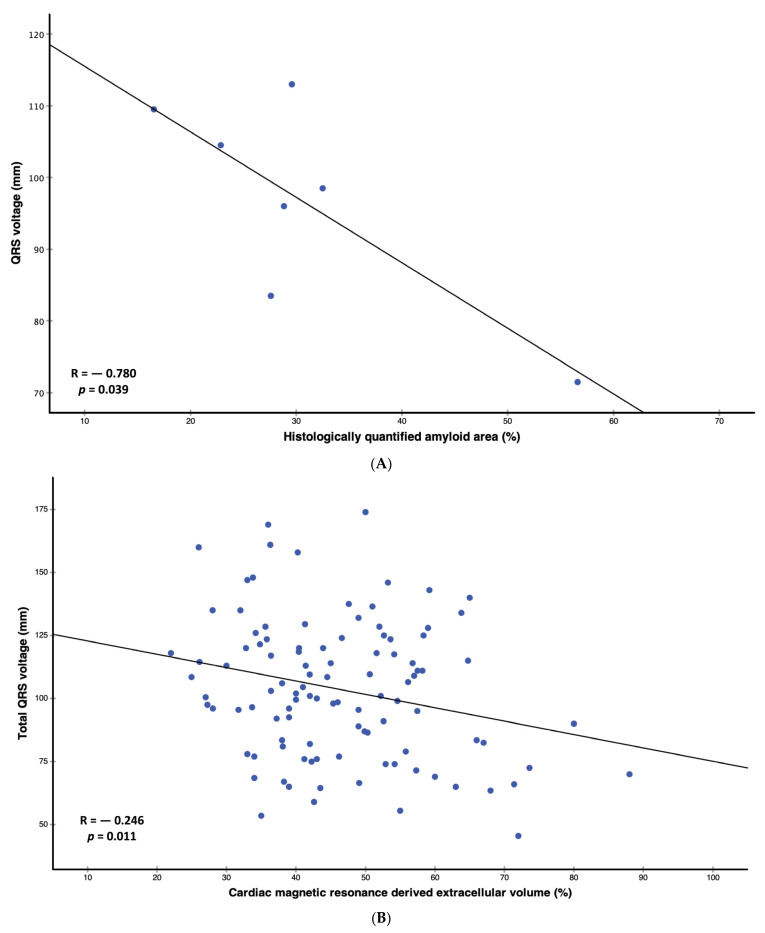
(**A**). Scatter plot of correlation analysis between histologically quantified amyloid area and total QRS voltage. In patients who underwent gold-standard invasive amyloid quantification using immunohistochemical staining of left ventricular endomyocardial biopsies (*n* = 7), we could detect a strong correlation of amyloid burden and total QRS voltage (R= −0.780, *p* = 0.039). (**B**). Scatter plot of correlation analysis between non-invasively quantified amyloid area and total QRS voltage. In patients who underwent non-invasive amyloid quantification using cardiac magnetic resonance T1 mapping (*n* = 105), we could detect a statistically significant correlation of amyloid burden and total QRS voltage (R = −0.246, *p* = 0.011).

**Figure 2 jcm-13-00368-f002:**
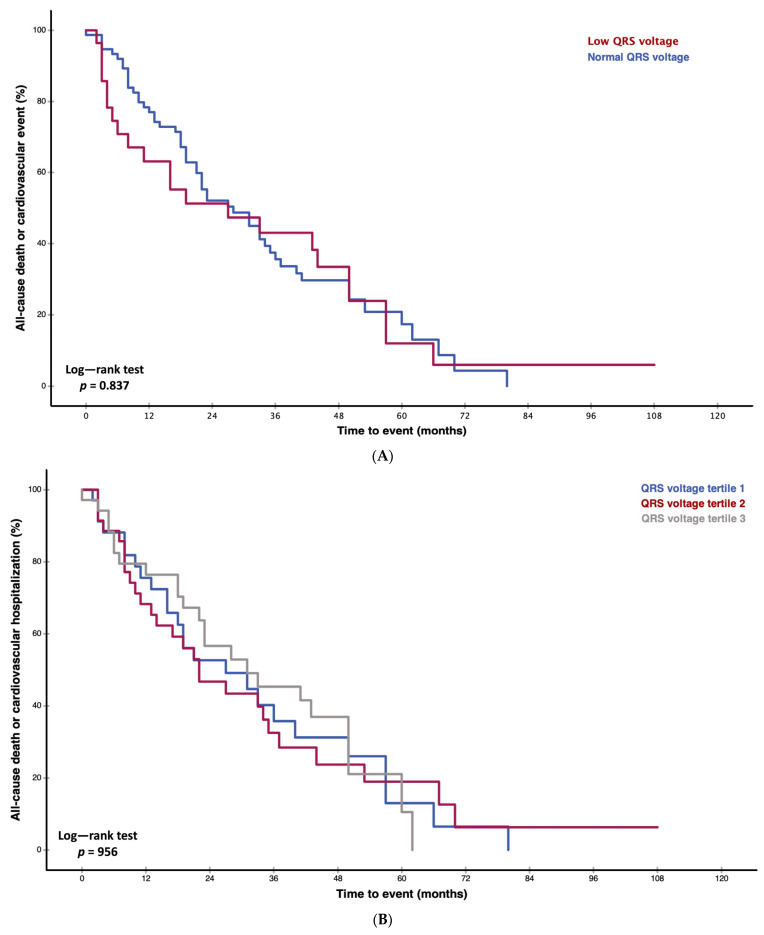
(**A**). Kaplan–Meier curves for cardiac amyloidosis cohort of 105 patients. With respect to the primary endpoint of all-cause death, heart transplantation, and cardiovascular-related hospitalizations, no statistical difference was detected (*p* = 0.837) when patients were stratified according to the presence of low voltage criteria (QRS amplitude in each precordial lead <1 mm or <5 mm in all peripheral leads). (**B**). Kaplan–Meier curves for cardiac amyloidosis cohort of 105 patients. With respect to the primary endpoint of all-cause death, heart transplantation, and cardiovascular-related hospitalizations, no statistical difference was detected (*p* = 0.956) when patients were stratified according total QRS voltage tertiles.

**Table 1 jcm-13-00368-t001:** Baseline characteristics of the cardiac amyloidosis cohort.

Variable	All-Comer Study Cohort *n* = 105
**Clinical parameters**	
Age, years (IQR)	77 (70–82)
Male sex, *n* (%)	78 (74.3)
Transthyretin amyloidosis, *n* (%)	87 (82.9)
Wild-type transthyretin amyloidosis, *n* (%)	78 (74.3)
Variant transthyretin amyloidosis, *n* (%)	9 (8.6)
Light chain amyloidosis, *n* (%)	18 (17.1)
Body mass index, kg/m^2^ (IQR)	27.0 (23.9–29.0)
New York Heart Association class ≥ III, *n* (%)	45 (42.9)
6-minute walk distance, m (IQR)	390 (285–473)
**Laboratory parameters**	
N-terminal pro brain natriuretic peptide, pg/mL (IQR)	2020 (849–3999)
Troponin t, ng/L (IQR)	43.0 (27.0–68.0)
Estimated glomerular filtration rate, mL/min/1.73 m^2^ (IQR)	54.6 (42.3–71.0)
National Amyloidosis Center stage I	49 (46.7)
National Amyloidosis Center stage II	24 (22.9)
National Amyloidosis Center stage III	14 (13.3)
**Histologically quantified amyloid area**	
Amyloid area, cm^2^ (IQR)	
**Electrocardiographic parameters**	
Total voltage, mm (IQR)	102 (81.5–123)
Total peripheral lead voltage, mm (IQR)	34.0 (23.8–41.8)
Total precordial lead voltage, mm (IQR)	67.0 (54.3–78.5)
Low voltage criteria, *n* (%)	29 (27.6)
Sinus rhythm, *n* (%)	66 (62.9)
Atrial fibrillation, *n* (%)	36 (34.3)
Atrial flutter, *n* (%)	3 (2.9)
Heart rate, bpm (IQR)	71 (62–82.5)
PR interval, ms (IQR)	195 (166–212)
QRS width, ms (IQR)	102 (86–134)
QT time, ms (IQR)	462 (438–484)
AV-Block 1°, *n* (%)	18 (17.1)
AV-Block 2° (Mobitz type 1), *n* (%)	1 (1.0)
Incomplete left bundle branch block, *n* (%)	7 (6.7)
Complete left bundle branch block, *n* (%)	17 (16.2)
Complete right bundle branch block, *n* (%)	8 (7.6)
Left anterior fascicular block, *n* (%)	8 (7.6)
Bifascicular block *n* (%)	9 (8.6)
Anterior pseudoinfarct pattern, *n* (%)	21 (20.0)
Ventricular premature complexes, *n* (%)	8 (7.6)
**Cardiac magnetic resonance imaging parameters**	
Extracellular volume, % (IQR)	43.9 (36.4–54.4)
LVEDD, mm (IQR)	44.0 (40.0–48.0)
RVEDD, mm (IQR)	39.0 (35.0–44.5)
Interventricular septum, mm (IQR)	18.0 (15.0–21.0)
Left atrial area, cm^2^ (IQR)	31.0 (27.0–35.0)
Right atrial area, cm^2^ (IQR)	28.0 (23.0–35.5)
Ascending aorta, mm (IQR)	36.0 (33.0–39.0)
Pulmonary artery, mm (IQR)	28.0 (25.0–31.0)
Left ventricular ejection fraction, % (IQR)	54.3 (45.0–61.3)
Left ventricular end-diastolic volume, mL (IQR)	157 (127–189)
Left ventricular cardiac output, L/min (IQR)	5.3 (4.6–6.4)
Left ventricular mass, g (IQR)	177 (141–211)
Right ventricular ejection fraction, % (IQR)	48.0 (41.0–56.5)
Right ventricular end-diastolic volume, mL (IQR)	165 (136–203)
Right ventricular cardiac output, L/min (IQR)	5.1 (4.5–6.2)

IQR indicates interquartile range. Bold indicates statistical significance.

**Table 2 jcm-13-00368-t002:** Correlation analysis for total myocardial voltage and clinical, laboratory, electrocardiographical, and cardiac imaging parameters.

Variable *n* = 105	Correlation Coefficient	*p* Value
**Clinical and laboratory parameters**		
Body mass index	−0.132	0.179
New York Heart Association class ≥III	−0.111	0.273
6-minute walk distance	0.092	0.460
Log N-terminal pro brain natriuretic peptide	−0.113	0.352
**Troponin T**	−0.221	**0.028**
Estimated glomerular filtration rate	0.106	0.280
National Amyloidosis Center stage I–III	0.014	0.894
**Electrocardiographical parameters**		
Heart rate	−0.118	0.231
PR interval	−0.104	0.408
**QRS width**	0.239	**0.014**
QT time	0.154	0.119
**Histologically quantified amyloid area**		
**Amyloid area, *n* = 7**	−0.780	**0.039**
**Cardiac magnetic resonance imaging parameters**		
**Extracellular volume**	−0.246	**0.011**
Left ventricular end-diastolic diameter	−0.103	0.294
Right ventricular end-diastolic diameter	−0.091	0.356
**Interventricular septum**	0.282	**0.004**
**Left atrial area**	0.229	**0.019**
Right atrial area	−0.033	0.738
Ascending aorta	0.051	0.605
Pulmonary artery	−0.061	0.539
Left ventricular ejection fraction	0.026	0.795
Left ventricular end-diastolic volume	0.104	0.2090
Left ventricular cardiac output	0.000	0.996
Left ventricular mass	0.174	0.105
Right ventricular ejection fraction	0.063	0.525
Right ventricular end-diastolic volume	−0.021	0.835
Right ventricular cardiac output	−0.027	0.784

Bold indicates statistical significance.

**Table 3 jcm-13-00368-t003:** Uni- and multivariable linear regression analyses between total QRS voltage and clinical, laboratory, electrocardiographical, as well as cardiac imaging parameters.

Variable	Beta	95% Confidence Interval	*p* Value	Adjusted Beta *	95% Confidence Interval	*p* Value
**Total voltage**						
Body mass index	−0.966	−2.381–0.450	0.179	−1.092	−2.510–0.326	0.130
New York Heart Association class ≥III	−8.781	−19.780–2.217	0.116	−6.548	−18.178–5.082	0.266
6-minute walk distance	0.018	−0.030–0.066	0.460	0.005	−0.051–0.062	0.857
Log N-terminal pro brain natriuretic peptide	−2.721	−7.398–1.956	0.251	−1.129	−6.002–3.745	0.647
Troponin T	−0.136	−0.257–−0.015	**0.028**	−0.115	−0.240–0.011	0.073
Estimated glomerular filtration rate	0.129	−0.107–0.366	0.280	0.051	−0.196–0.299	0.681
National Amyloidosis Center stage I-III	−0.204	−8.015–7.607	0.959	0.402	−7.562–8.365	0.920
**Electrocardiographical parameters**						
Heart rate	−0.230	−0.607–0.148	0.231	−0.239	−0.615–0.138	0.211
PR interval	−0.086	−0.292–0.120	0.408	−0.114	−0.341–0.113	0.319
**QRS width**	0.218	0.045–0.391	**0.014**	0.227	0.048–0.406	**0.013**
QT time	0.128	−0.033–0.289	0.119	0.130	−0.033–0.293	0.117
Amyloid area	−0.913	−1.756–−0.070	**0.039**	0.900	−8.675–10.476	0.725
**Extracellular volume**	−0.530	−0.939–−0.122	**0.011**	−0.610	−1.017–−0.204	**0.004**
Left ventricular end-diastolic diameter	−0.427	−1.229–0.375	0.294	−0.454	−1.268–0.360	0.271
Right ventricular end-diastolic diameter	−0.329	−1.033–0.375	0.356	−0.368	−1.082–0.346	0.309
**Interventricular septum**	1.901	0.637–3.165	**0.004**	1.749	0.403–3.096	**0.011**
**Left atrial area**	0.767	0.128–1.405	**0.019**	0.793	0.160–1.426	**0.015**
Right atrial area	−0.119	−0.823–0.585	0.738	−0.193	−0.956–0.570	0.616
Ascending aorta	0.335	−0.945–1.614	0.605	0.401	−0.988–1.790	0.568
Pulmonary artery	-0.350	−1.475–0.776	0.539	−0.002	−1.203–1.199	0.997
Left ventricular ejection fraction	0.058	−0.387–0.503	0.795	0.149	−0.301–0.600	0.511
Left ventricular end-diastolic volume	0.062	−0.054–0.178	0.290	0.053	−0.077–0.183	0.419
Left ventricular cardiac output	0.009	−3.824–3.842	0.996	1.009	−3.040–5.058	0.622
**Left ventricular mass**	0.091	−0.019–0.201	0.105	0.129	0.002–0.256	**0.046**
Right ventricular ejection fraction	0.156	−0.329–0.640	0.525	0.242	−0.263–0.746	0.344
Right ventricular end-diastolic volume	−0.011	−0.119–0.097	0.835	−0.024	−0.143–0.096	0.694
Right ventricular cardiac output	−0.500	−4.109–3.109	0.784	0.070	−3.610–3.751	0.970

Bold indicates statistical significance. * indicates adjustment for type of amyloid (ATTR/AL), sex, and body mass index.

**Table 4 jcm-13-00368-t004:** Electrocardiographic characteristics of patients stratified by extracellular volume.

Variable	ECV < Median (43.9%)	ECV ≥ Median (43.9%)	*p* Value
**Electrocardiographic parameters**			
Total voltage, mm (IQR)	103 (82.8–121)	100 (77.5–124)	0.495
Total peripheral lead voltage, mm (IQR)	34.5 (24.3)	33.8 (22.6–41.0)	0.453
Total precordial lead voltage, mm (IQR)	68.0 (55.8–80.0)	63.8 (52.6–76.6)	0.624
Low voltage criteria, *n* (%)	12 (22.6)	17 (32.7)	0.249
Sinus rhythm, *n* (%)	37 (69.8)	29 (55.8)	0.137
Atrial fibrillation, *n* (%)	14 (26.4)	22 (42.3)	0.086
Atrial flutter, *n* (%)	2 (3.8)	1 (1.9)	0.569
Heart rate, bpm (IQR)	69.0 (60.5–80.0)	71.0 (64.8–86.0)	0.115
**PR interval, ms (IQR)**	180 (157–204)	204 (171–218)	**0.016**
QRS width, ms (IQR)	96.0 (86.0–135)	108 (92.0–132)	0.255
QT time, ms (IQR)	454 (431–481)	466 (451–489)	0.079
Any AV-Block, *n* (%)	7 (13.2)	12 (23.1)	0.189
AV-Block 1°, *n* (%)	7 (13.2)	11 (21.2)	0.134
AV-Block 2° (Mobitz type 1), *n* (%)	0 (0.0)	1 (1.9)	0.310
Any bundle branch block, *n* (%)	22 (41.5)	27 (23.1)	0.285
Incomplete left bundle branch block, *n* (%)	3 (5.7)	4 (7.7)	0.676
Complete left bundle branch block, *n* (%)	12 (22.6)	5 (9.6)	0.070
Complete right bundle branch block, *n* (%)	2 (3.8)	6 (11.5)	0.134
**Left anterior fascicular block, *n* (%)**	1 (1.9)	7 (13.5)	**0.025**
Bifascicular block *n* (%)	4 (7.5)	5 (9.6)	0.705
Anterior pseudoinfarct pattern, *n* (%)	9 (17.0)	12 (23.1)	0.435
Ventricular premature complexes, *n* (%)	3 (5.7)	5 (9.6)	0.445

IQR indicates interquartile range. Bold indicates statistical significance.

## Data Availability

The raw data supporting the conclusions of this article will be made available by the authors on reasonable request.

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
