# Peer review of "Amyloid Burden Correlates with Electrocardiographic Findings in Patients with Cardiac Amyloidosis—Insights from Histology and Cardiac Magnetic Resonance Imaging"

_jcm, 2024, doi:10.3390/jcm13020368_

Round 1

Reviewer 1 Report

Comments and Suggestions for Authors

Dear authors,

I am writing to express my sincere appreciation for your insightful article, "Amyloid Burden Correlates with Electrocardiographic Findings in Patients with Cardiac Amyloidosis - Insights from Histology and Cardiac Magnetic Resonance Imaging." Your research has significantly contributed to our understanding of the intricate relationship between amyloid burden and electrocardiographic manifestations in patients with cardiac amyloidosis.

The meticulous integration of histological analysis and cardiac magnetic resonance imaging (MRI) in your study represents a commendable approach. By combining these methodologies, your research provides a comprehensive perspective, offering valuable insights into the underlying mechanisms that link amyloid deposition and cardiac electrophysiology. This interdisciplinary approach not only enhances the scientific rigor of your work but also opens avenues for future research and clinical applications. I see this single center research with close follow up of these cases as strength that validates your results.

Furthermore, the clarity and coherence of your writing make the complex subject matter accessible to a broad audience, including both clinicians and researchers. The precise articulation of your findings and their implications demonstrates a commitment to effective communication, fostering a deeper understanding of the importance of amyloid burden in the context of cardiac health.

Your findings represent a significant contribution to the existing body of knowledge, as we strive to improve diagnostic and therapeutic strategies for cardiac amyloidosis. Your work serves as guidance, directing future research endeavors and clinical interventions, and enriches our understanding of cardiac amyloidosis, but also inspires further exploration into the intricate interplay between molecular pathology and clinical outcomes in cardiovascular diseases.

Minor revision required:

The abstract exceeds 200 words – please correct

Abstract Background section – please change he word “role” into “effect/impact” that seem more suitable

Abstract Results - lower case “wild-type Transthyretin”

Introduction section  - please merge the first two sentences into one, the second one has no verb.

-  the second sentence of the second paragraph – please rephrase for clarity

Methods - Setting and study design – first sentence please rephrase for clarity

Methods - Diagnosis of cardiac light chain amyloidosis – current recommendations – please specify them

-   HistoQuest software (TissueGnostics®, Vienna Austria) please describe how amyloid quanification is done

Discussion section - “Moreover, QRS voltage also correlated with ECV (n=105, R= -0.246) which represents the gold-standard for non-invasive amyloid quantification” – please add references

Please verify and correct references for “Prognostic significance of electrocardiographic findings in cardiac amyloidosis”

Author Response

Please be referred to the attched file. 

Kind regards. 

Reviewer 2 Report

Comments and Suggestions for Authors

I find the article very original; there are few similar ones, especially using the non-invasive algorithm for cardiac amyloidosis. There are some easily correctable errors, which in my opinion are minor, and their correction would improve the article.

Corrections:

Abstract:

  • The concluding phrase, "additional pathophysiological mechanism…," is not supported by the discussed results. Either complete the results section or delete that phrase.

Methodology:

  • After "University of Vienna," change the italics of "and."

  • Why were patients with a BMI greater than or equal to 35 excluded?

  • If genetic testing was recommended for all patients, why are the results not included? I strongly recommend including it, as it could provide new information.

  • The "outcome measures" section needs to be rewritten. Throughout, it is mentioned that the explored finding is the relationship between magnetic resonance and ECG findings, but it is not discussed as the primary endpoint. This section or the entire article should be rewritten.

  • SPSS and STATA are software; please cite the companies that develop them and add the copyright symbol.

Results:

  • The origin of the patients is not mentioned—whether they come from an outpatient clinic, hospitalization, Cardiology, Internal Medicine, etc.
  • Are these consecutive patients? Specify in the text. It has implications for selection bias.
  • When discussing correlations, although significant, all are weak except QRS with quantified amyloid area; specify in the text.

Discussion:

  • Specify the possible selection bias that patients with less expressive ECGs may not have been included due to not being detected.

Author Response

Please be referred to the attached file. 

Kind regards. 
